# Federated Reinforcement Learning for Training Control Policies on Multiple IoT Devices

**DOI:** 10.3390/s20051359

**Published:** 2020-03-02

**Authors:** Hyun-Kyo Lim, Ju-Bong Kim, Joo-Seong Heo, Youn-Hee Han

**Affiliations:** 1Department of Interdisciplinary Program in Creative Engineering, Korea University of Technology and Education, Cheonan 31253, Korea; glenn89@koreatech.ac.kr (H.-K.L.); chil1207@koreatech.ac.kr (J.-S.H.); 2Department of Computer Science Engineering, Korea University of Technology and Education, Cheonan 31253, Korea; rlawnqhd@koreatech.ac.kr

**Keywords:** Actor–Critic PPO, federated reinforcement learning, multi-device control

## Abstract

Reinforcement learning has recently been studied in various fields and also used to optimally control IoT devices supporting the expansion of Internet connection beyond the usual standard devices. In this paper, we try to allow multiple reinforcement learning agents to learn optimal control policy on their own IoT devices of the same type but with slightly different dynamics. For such multiple IoT devices, there is no guarantee that an agent who interacts only with one IoT device and learns the optimal control policy will also control another IoT device well. Therefore, we may need to apply independent reinforcement learning to each IoT device individually, which requires a costly or time-consuming effort. To solve this problem, we propose a new federated reinforcement learning architecture where each agent working on its independent IoT device shares their learning experience (i.e., the gradient of loss function) with each other, and transfers a mature policy model parameters into other agents. They accelerate its learning process by using mature parameters. We incorporate the actor–critic proximal policy optimization (Actor–Critic PPO) algorithm into each agent in the proposed collaborative architecture and propose an efficient procedure for the gradient sharing and the model transfer. Using multiple rotary inverted pendulum devices interconnected via a network switch, we demonstrate that the proposed federated reinforcement learning scheme can effectively facilitate the learning process for multiple IoT devices and that the learning speed can be faster if more agents are involved.

## 1. Introduction

Recently, reinforcement learning has been applied to various fields and shows better performance than humans. Reinforcement learning [1] is how the agent observes the simulation or real environment and chooses an action that maximizes the cumulative future reward. In particular, after the development of Deep Q-Network (DQN) by Google DeepMind, reinforcement learning has been applied to Atari Games in 2015 [2], Go in 2016 and in 2018 (AlphaGo and AlphaZero) [3,4], and StarCraft 2 in 2019 (AlphaStar) [5]. As a result, the optimal control using reinforcement learning has been studied mostly in the area of game playing.

On the other hand, the real-world control systems, such as inverted pendulums, robot arms, quadcopters, continuum manipulators, chemical reactors, and so on, are of different categories of nonlinearities. One of the most effective ways to develop a nonlinear control strategy, reinforcement learning-based control schemes interact directly with these systems in a trial-and-error manner to train agents to learn optimal control policies [6,7,8,9]. In our previous work [10], we also proved that a DQN agent can control a rotary inverted pendulum (RIP) successfully, even though the agent is deployed remotely from the pendulum. An imitation learning approach was applied to reduce the learning time and mitigate the instability problem in the previous study.

Multiple IoT devices of one type are often installed and utilized simultaneously in a domain. For example, in a factory that produces a large number of cars per day, many robotic arms of the same type assemble them precisely. In order to control such multiple robotic arms optimally, a reinforcement learning agent that controls one IoT device can collaborate with other agents that control the other IoT devices in order to learn its optimal control policy faster. Such a collaboration is able to accelerate the learning process and mitigate the instability problem for training multiple IoT devices.

Although multiple IoT devices of the same type are produced on the same manufacturing line, their physical dynamics are usually somewhat different. Therefore, there is no guarantee that an agent who interacts only with one IoT device and learns the optimal control policy will control another IoT device well. A simple copy of a mature control policy model to multiple devices cannot be a solution, but a kind of cooperative method is required to train multiple IoT devices efficiently. We consider that the cooperative reinforcement learning agents sharing learning outcomes can have more generalization capability than an agent that interacts only one IoT device.

Bonawitz et al. [11] proposed a new machine learning system based on federated learning, which is a distributed deep learning approach to enable training on a large corpus of decentralized data residing in multiple IoT devices like a mobile phone. The federated reinforcement learning enables learning to be shared among other agents on IoT devices by conducting learning in separate environments through distributed multi-agents and collecting learning experiences through a broker. In this paper, we propose a new federated reinforcement learning scheme for controlling multiple IoT devices of the same type by borrowing the idea of the collaborative approach of the federated learning.

In the proposed scheme, each agent on multiple IoT devices shares its learning experiences, i.e., a gradient of loss function computed on each device, with each other. This strategy enhances the generalization ability of each agent. Each agent also transfers its deep learning model parameters into other agents when its learning process is finished, so that other workers can accelerate its learning process by utilizing the mature parameters. Therefore, the proposed federated learning scheme consists of two phases: (1) gradient sharing phase and (2) transfer learning phase. Inside the overall procedure of the proposed scheme, the actor–critic proximal policy optimization (Actor–Critic PPO) [12,13,14] is incorporated instead of DQN. Actor–Critic PPO consists of the policy neural network (actor) that determines the optimal behavior of the agent, and the value neural network (critic) that serves to evaluate the policy. It is inspired by TRPO [15], but is known to be simpler and provide superior performance in many domains [7]. We evaluate the performance of our federated reinforcement learning scheme incorporating Actor–Critic PPO with the three RIP devices.

The contributions of this paper can be summarized as follows:We propose a new federated reinforcement learning scheme to allow multiple agents to control their own devices of the same type but with slightly different dynamics.We verify that the proposed scheme can expedite the learning process overall when the control policies are trained for the multiple devices.

The remainder of this paper is organized as follows. In Section 2, we review related studies and state the motivation for our work. In Section 3, we describe the overall system architecture and procedure of the proposed federated reinforcement learning scheme. In Section 4, we explain the operation methods of the gradient sharing and model transfer, and present the details of the Actor–Critic PPO algorithm used for reinforcement learning. In Section 5, we prove the effectiveness of the proposed scheme by applying it to three real RIP devices. Finally, we provide concluding remarks and future work in Section 6.

## 2. Related Work

### 2.1. Federated Reinforcement Learning

Federated learning is a distributed machine learning technique that trains an algorithm across multiple decentralized edge devices holding local data samples, without exchanging their data samples [11,16,17,18]. It allows multiple edge devices to jointly learn the deep learning model of local data without disclosing data to a central server or other devices. This type of privacy collaboration is accomplished by a simple three-step process. In the first step, all devices bring the latest global model from the server. Next, the device uses a stochastic gradient descent (SGD) algorithm based on local training data to update the global model imported from the server to the local model. Finally, to form a new global model, all devices collect and integrate improved local models and upload them back to the server. These steps are repeated until a certain convergence criterion is satisfied, or lasted for a long period to improve the deep learning model continuously.

Transfer learning is a machine learning technique that focuses on storing knowledge gained while solving one problem and applying it to a different but related problem [19]. Glatt et al. [20] proposed a method of applying transfer learning to reinforcement learning. They made deep learning models to control some Atari games with DQN. To train other Atari games, then, the pre-trained DQN models were transferred to the model of the new DQN. By applying transfer learning to reinforcement learning, the learning process is greatly accelerated. Also, in 2015, Google DeepMind worked on extending DQN to a distributed architecture [21]. In the architecture, the parameter server collects each gradient through the DQN agents and updates the global model. The parameter server copies the updated global model to each agent’s target network to share experiences among distributed agents. The use of distributed DQN architecture improves learning time and performance. There have been many researches on a distributed architecture for transfer learning and gradient sharing in reinforcement learning.

The combination of federated learning and reinforcement learning, namely federated reinforcement learning (FRL), was first studied in [22]. In the study, the authors demonstrated that the FRL approach is capable of making full use of the joint observations (or states) from an environment, and outperforms a simple DQN with the partial observation of the same environment. FRL was also applied to autonomous driving [23], where all the participant agents make steering control actions with the knowledge learned by others, even when they are acting in very different environments. Even in robot system control, FRL was used to make robot agent models fuse and transferred their experience so that they can effectively use prior knowledge and quickly adapt to new environments [24]. However, the previous FRL schemes were evaluated and verified in software such as games or simulation rather than real devices. In addition, Actor–Critic PPO has not yet been applied to FRL research.

In this paper, we propose a new scheme to apply the federation method to reinforcement learning using gradient sharing and transfer learning. Unlike the previous study, the proposed method is used to train multiple real IoT devices simultaneously in a distributed architecture rather than virtual environments.

### 2.2. Actor–Critic PPO

Policy-based reinforcement learning algorithms directly optimize the agent’s policy and have shown excellent performance in various fields. In particular, PPO algorithm [12] has been proposed to mitigate the disadvantage of the policy-based algorithm, where the policy can be changed drastically in the process of gradually updating it. The PPO agent samples data through interaction with the given environment and optimizes its objective function using an SGD optimization algorithm. In a model optimization, there are multiple iterations of model parameter updates with data generated from the environment. That is, the importance sampling technique is used to take advantage of past experiences for sample efficiency. This feature is well suited for learning how to directly control IoT devices, since acquiring rich training data from real IoT devices is time-consuming and very expensive. In addition to that, the drastic policy change is mitigated due to the proximal trust region which is constrained through the clipping method applied to the objective function.

The Actor–Critic method [13,25] has been also known as a reinforcement learning framework to lead a good learning performance. It uses two deep learning models, one called actor model and the other called critic model (sometimes, the hidden layers are shared by actor and critic models since the parameters useful for estimating the value function could also be useful for selecting actions. In this study, the two models share the hidden layers, too). The actor model performs the task of learning what action to be selected under a particular observation of the environment (i.e., the control policy). When the action selected by the actor model is performed, the agent gets a reward from the environment. This reward is taken in by the critic model. The role of the critic model is to learn to evaluate if the action taken by the actor model led the environment to be in a better state or not, and its feedback is used to the actor model optimization. It outputs a real number indicating a rating of the action taken in the previous state. By comparing this rating, the agent can compare its current policy with a new policy and decide how it improves the actor model to take better actions. In this paper, we set up this actor–critic framework into each agent working on its independent IoT device and implement the PPO algorithm into the framework.

In our previous work [10], we implemented the DQN algorithm in a single RIP device, Quansar QUBE-Servo 2 [26]. In this paper, we extend the previous work and propose a new federated reinforcement learning scheme that can make each agent on multiple IoT devices learn the optimal control policy collaboratively. The proposed federated reinforcement learning scheme, unlike the existing FRL schemes, applies two strategies, (1) gradient sharing and (2) model transfer, to allow distributed IoT devices to share the learning experience. In this way, the proposed federated reinforcement learning speeds up the learning process and improves the generalization capability that allows multiple agents to control their own IoT devices robustly.

## 3. System Architecture & Overall Procedure

In this section, we describe the proposed federated reinforcement learning system architecture and overall procedure to allow multiple agents to control their own IoT devices. The proposed system consists of one chief node and *N* worker nodes. The workers have their own independent environment (i.e., IoT device) and train its actor and critic models to control the environment optimally through its reinforcement algorithm. On the other hand, the chief node mediates the federated work across the *N* workers and ensures that each worker’s learning process is synchronized.

The reinforcement learning has been mostly researched in simulation environments to consist of software. In the simulation environment, most elements of the device are placed in cyberspace and implemented in software. However, as shown in Figure 1, the manufactured real IoT devices are positioned in the physical space and the worker containing our reinforcement learning agent is usually placed in cyberspace because of the IoT device’s constrained resource. Therefore, the elements of cyberspace and physical space interact in real-time is important. Besides, it is vital that the reinforcement learning agent must not be broadly changed to control the real IoT device. In other words, the control task in the equivalent process irrespective of whether the environment to be controlled exists in the cyber or physical space should be performed by the reinforcement learning agent. We allow workers to set up a cyber environment to correspond with the physical environment, so that the state of the real IoT device can be observed and be controlled by the reinforcement learning agents only over the cyber environment. When observing and controlling the real IoT device, the reinforcement learning agent may not need to know that the real device is placed in the physical space.

Figure 2 shows the system overall procedure for the proposed federated reinforcement learning scheme. The process where each worker sends and receives messages back to the chief is called *Round*. For a round, each worker starts a sequential interaction (i.e., an episode) with its environment at the time step *t* = 0, and finishes at the time step *T* when the episode end condition is met (the episode end conditions vary according to the type of control IoT devices and the control objective). At every time step *t*, the worker receives a representation $s_{t}$ of the environment’s state and selects an action $a_{t}$ that is executed in the environment which in turn provides a reward signal $r_{t+1}$ and a representation $s_{t+1}$ of the successor state. For every time step *t*, the worker stores the tuple $\;<\;s_{t},a_{t},r_{t+1},s_{t+1}\;>\;$ as its experience into its trajectory memory. The tuples of trajectory memory are continuously maintained across every round. But, the size of trajectory memory is limited, so that the tuples are inserted and removed according to an organizing and manipulating rule, such as the first-in first-out (FIFO) rule.

In each round, each worker’s reinforcement learning algorithm calculates the gradients for the optimization of actor and critic models by using the tuples stored in the trajectory memory. For the deep learning models of a worker, the gradients are the vectors of partial derivatives with respect to the parameters of the models and they are used to find the optimal models to control the IoT device.

The two phases, (1) gradient sharing and (2) model transfer, are synchronized and mediated by the chief node. In the gradient sharing phase, the chief collects the actor model’s gradient produced by the learning process of workers (step ① in Figure 2), averages them (step ②), and sends it back to the workers (step ③). The workers optimize the current actor model once more using the average gradient received from the chief. In our architecture, the gradient represents the experience of the learning task that a worker has made to control its IoT device. That is, workers share their experience with each other during the gradient sharing phase.

The idea of gradient sharing was first introduced by [27]. However, it has been used asynchronously for the parallel or distributed SGD implementation, whereas we utilize it in a synchronous manner. Even though the asynchronous gradient sharing can minimize worker idle time, it is not recommended due to the added noise from stale gradients (referred to as the “delayed gradient problem”) [28,29]. With synchronous gradient sharing, the chief waits for all gradients to be available before computing the average and sending it back to the workers. The drawback is that some workers may be slower than others, so other workers may have to wait for them at every round. To reduce the waiting time, we could ignore the gradients from the slowest few workers (typically 10%).

After multiple rounds of the gradient sharing, a worker comes to satisfy the predefined criteria for completing the learning process. At this time, the next round is executed for the model transfer phase. During the model transfer phase, the worker completing its learning process sends its mature actor model parameters to the chief (step ④ in Figure 2). When receiving the mature actor model parameters from a worker, the chief considers them the best ones, and it transfers them to the rest N−1 workers (step ⑤). And then, the N−1 workers replace their model parameters with the mature model parameters, so that the leaning time can be reduced for the workers with slow learning.

Although the mature actor model is incorporated to the N−1 workers, several gradient sharing phases may be still needed among the workers, since the inherent dynamics of each IoT device are slightly different from each other. Therefore, the same procedure of the steps from ① to ⑤ in Figure 2 is performed repeatedly between the rest of the workers. By continuously performing this procedure, the learning processes of all workers are completed when the predefined condition is satisfied at the last worker.

## 4. Federated Reinforcement Learning Algorithm

In our study, each worker conducts individual training with the Actor–Critic PPO algorithm on an independent IoT device. The key advantage of Actor–Critic PPO is that a new update of the policy model does not change it too much from the previous policy. It leads to less variance in learning, but ensures smoother policy update and also ensure that the worker does not go down an unrecoverable path of taking senseless actions. This feature is particularly important for optimal IoT device control because optimal control learning for IoT devices in the physical environment takes more time than software in the cyber environment.

An actor model (i.e., policy model) $\pi_{\theta }$ has its own model parameters θ. With an actor model, a worker performs the task of learning what action to take under a particular observed state of the IoT device. The worker sends the action predicted by the actor model to the IoT device and observes what happens in the IoT device. If something positive happens as a result of the action, then a positive response is sent back in the form of a reward. If a negative occurs due to the taken action, then the worker gets a negative reward. This reward is taken in by the critic model $V_{\mu }$ with its model parameter μ. The role of the critic model inside a worker is to learn to evaluate if the action taken by the actor model led the IoT device to be in a better state or not, and the critic model’s feedback is used to the actor model optimization.

Figure 3 shows the overall process of an Actor–Critic PPO algorithm conducted in a worker whenever every episode ends. First, the Actor–Critic PPO obtains a finite mini-batch of sequential samples (i.e., experience tuples) from the trajectory memory. In a mini-batch, the time step ($t_{0}$ in Figure 3) of starting tuple is chosen at random, but all subsequent tuples in the mini batch must be continuous.

In a general policy gradient reinforcement learning, the objective function $L^{P}$ is as follows:(1)$$L^{P}\left(\theta \right)=\hat{E}\left[\log\pi_{\theta }\left(a_{t}|s_{t}\right){\hat{A}}_{t}\right]$$
where $\hat{E}\left[\ldots \right]$ is the empirical average over a finite batch of samples, and ${\hat{A}}_{t}$ is an estimator of the advantage function at time step *t*. With the discount factor γ∈[0,1], we use the generalized advantage estimator (GAE) [30] to calculate ${\hat{A}}_{t}$. The GAE is
(2)$${\hat{A}}_{t}=\delta_{t}+\left(\gamma \lambda \right)\delta_{t+1}^{V}+{\left(\gamma \lambda \right)}^{2}\delta_{t+2}^{V}\cdots {\left(\gamma \lambda \right)}^{U-t+1}\delta_{U-1}^{V}$$
where λ is the GAE parameter (λ∈[0,1]), *U* is the sampled mini-batch size, and $\delta_{t}=r_{t}+\gamma V_{\mu }\left(s_{t+1}\right)-V_{\mu }\left(s_{t}\right)$. The objective (i.e., loss) function $L^{V}$ is as follows:(3)$$L^{V}\left(\mu \right)=\hat{E}\left[L_{t}^{V}\left(\mu \right)\right]=\hat{E}\left[|{\hat{V}}_{\mu }^{target}\left(s_{t}\right)-V_{\mu }\left(s_{t}\right)|\right]$$
where the target value of time-difference error (TD-Error) ${\hat{V}}_{\mu }^{target}\left(s_{t}\right)=r_{t+1}+\gamma V_{\mu }\left(s_{t+1}\right)$. The parameters of $V_{\mu }$ are updated by an SGD algorithm (i.e., Adam [31]) with the gradient $\nabla L^{V}$:(4)$$\mu =\mu -\eta_{\mu }\nabla L^{V}\left(\mu \right)$$
where $\eta_{\mu }$ is the learning rate for the critic model optimization.

In the actor model of TRPO and PPO, instead of the objective function presented in Equation (Equation 1), the worker uses the importance sampling to obtain the expectation of samples gathered from an old policy $\pi_{\theta_{old}}$ under the new policy $\pi_{\theta }$ we want to refine. They maximize the following surrogate objective function $L^{CPI}$:(5)$$L^{CPI}\left(\theta \right)=\hat{E}\left[\frac{\pi_{\theta }\left(a_{t}|s_{t}\right)}{\pi_{\theta_{old}}\left(a_{t}|s_{t}\right)}{\hat{A}}_{t}\right].$$

With a small value δ, TRPO optimizes $L^{CPI}$ subject to the constraint
$$\hat{E}\left[KL\left[\pi_{\theta_{old}}\left(\cdot |s_{t}\right),\pi_{\theta }\left(\cdot |s_{t}\right)\right]\right]\leq \delta$$
on the amount of the policy update. KL indicates the Kullback–Leibler divergence [32]. PPO inherits the benefit from TRPO, but it is much simpler to implement, allows multiple optimization iterations, and empirically presents a better sample efficiency than TRPO. With the probability ratio $R_{t}\left(\theta \right)=\frac{\pi_{\theta }\left(a_{t}|s_{t}\right)}{\pi_{\theta_{old}}\left(a_{t}|s_{t}\right)}$, the PPO objective function $L^{CLIP}$ is given by
(6)$$\begin{array}{cc} L^{CLIP}\left(\theta \right) & =\hat{E}\left[L_{t}^{CLIP}\left(\mu \right)\right]\\ \; & =\hat{E}\left[min\left(R_{t}\left(\theta \right),clip\left(R_{t}\left(\theta \right),1-\epsilon ,1+\epsilon \right)\right){\hat{A}}_{t}\right] \end{array}$$
where ϵ is the clipping parameter. The clipped objective function $L^{CLIP}$ does not makes PPO greedy in favoring actions with positive advantage, and much quick to avoid actions with a negative advantage function from a mini-batch of samples. The parameters of $\pi_{\theta }$ are updated by an SGD algorithm with the gradient $\nabla L^{CLIP}$ for the negative of the clipped objective function (i.e., $-L^{CLIP}$):(7)$$\theta =\theta -\eta_{\theta }\nabla L^{CLIP}\left(\theta \right)$$
where $\eta_{\theta }$ is the learning rate for the actor model optimization.


With Actor–Critic PPO, our federated reinforcement learning algorithm is provided in Algorithms 1 and 2. The chief and all workers share the parameter *M* which indicates the maximum number of rounds (i.e., episodes). The chief maintains the set of all workers *W*. Whenever a chief receives the actor model parameter $\theta_{w}$ from a worker *w*, the chief removes it from *W* at the end of round. If *W* is empty, the chief finishes its task. In a worker, *K* is the number of model optimizations in one round. The multiple model optimizations are conducted to further improve sample efficiency.

## 5. Experiments

In this section, we apply the proposed federated reinforcement learning scheme to the real IoT devices. We validate the effectiveness of gradient sharing and transfer learning for our federated reinforcement learning in the real IoT devices. We also validate the effect of the number of workers on the performance of the proposed scheme.

**Algorithm 1:** Federated RL (Chief)

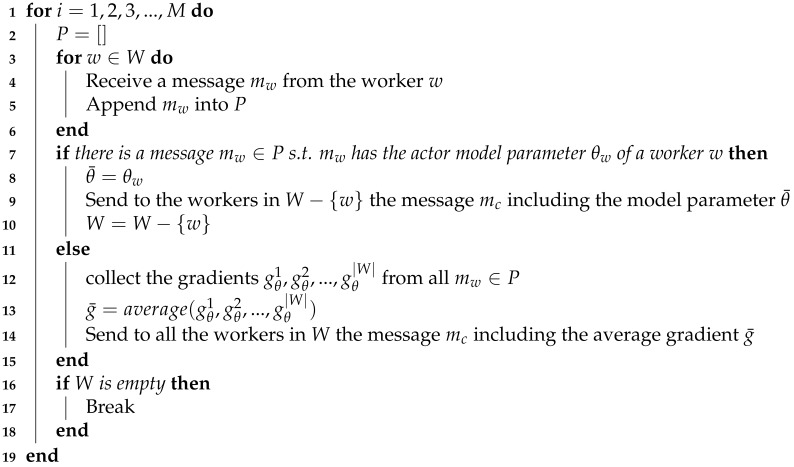



**Algorithm 2:** Federated RL (Worker *w*)

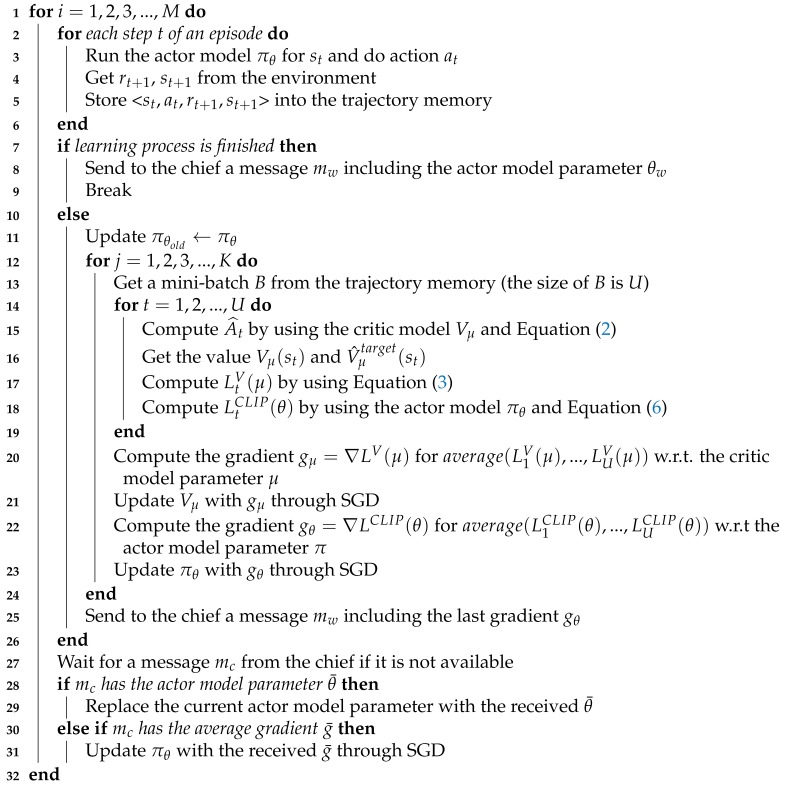



### 5.1. Experiment Configuration

As the real IoT device environment, we choose the RIP device (Quanser QUBE^TM^-Servo 2 [26]). It is a highly unstable nonlinear IoT device and has been used as a usual device in the nonlinear control engineering field.

The experimental system configuration for controlling the real multiple IoT devices optimally is shown in Figure 4. In the system, there are three workers and one chief, and each worker is connected with its RIP device. The RIP device is installed with an optical encoder that provides status information about each angular position and angular velocity of the pendulum and motor. The Raspberry Pi 3 Model B (Raspberry Pi) connects to RIP device and a worker through a serial peripheral interface (SPI). The Raspberry Pi takes the motor power index (i.e., control input) from the reinforcement learning agent in a worker, permutes it into a voltage value, and eventually sends it to the RIP device. Also, it takes observed state information (i.e., the angular position and angular velocity of the pendulum and the motor) from the RIP device, and sends them to the reinforcement learning agent in a worker. There is also a switch for the connection between the three workers and the chief. For the communication among the Raspberry Pi nodes, the workers, and the chief, the MQTT protocol is used.

The workers and the chief are deployed on Ubuntu 16.04 LTS. The proposed scheme using the Actor–Critic PPO algorithm is implemented using Python 3.6 and PyTorch 1.2. A multi-layer perceptron with the three hidden layers and two separate output layers is used to constitute the actor and critic models. The two models share the three hidden layers, where each layer includes 128 neurons. For the actor model, the first output layer yields three values (i.e., three types of actions) that sum to one. For the critic model, the second output layer yields a single value to evaluate the action selected by the actor model. The other hyper-parameters of the Actor–Critic PPO algorithm are listed in Table 1.

### 5.2. State, Action, and Reward Formulation

First, we perform an upswing control at the beginning of each round. This process forces the motor to move until the pendulum is facing upwards. This process has been empirically implemented using existing control models in the mechanical field. The proposed Actor–Critic PPO reinforcement learning algorithm then tries to balance the pendulum.

For the proposed algorithm, the state of reinforcement learning consists of observed information from the environment. The observed information is as follows: (1) pendulum angle, (2) pendulum angular velocity, (3) motor angle, and (4) motor angular velocity. In an episode, at every step, a new state is provided for the actor and critic models.

The action is selected by each worker’s actor model and the RIP device controls the motor by the selected action. The selected actions are −60, 0 and 60 as the power of the motor. These values are the power to turn the motor. If the selected action is negative 60, the direction of turning is from right to left, whereas the direction of rotation is positive 60, the direction of turning is from left to right. On the Raspberry Pi, the motor power index is changed to the motor voltage value, which is fed to the RIP device.

Designing rewards is important for reinforcement learning challenges. At each step, the reward is calculated after applying the selected action in the current state by the cyber environment. The value of reward depends on each step of success or failure. If the pendulum of the RIP is within a range of ±7.5 degrees (i.e., the pendulum is standing upright) of the step, the step succeeds and the reinforcement learning agent is rewarded with +1. If the step fails, the reinforcement learning agent is rewarded with 0 and the episode is terminated. When the episode is terminated, the score is the sum of the reward for all steps of the episode. If the weighted moving average of the reward in the last 10 episodes is 2450, then the learning is determined to be complete.

### 5.3. Effect of Gradient Sharing & Transfer Learning

Figure 5 shows the effectiveness of the proposed federated reinforcement learning for the experimental system. In particular, the advantages of gradient sharing and model transfer are illustrated in the figure. For each of the three workers, Figure 5a shows the change of the score and the loss values of the actor model when each worker performs the learning process individually, while Figure 5b shows the change of the ones when the proposed scheme is applied. As known in the two figures, at about 300 episodes, the learning process of all workers is completed when the proposed federated learning scheme is applied. However, without the proposed scheme, the learning process of Worker I and II are completed at about 820 episodes and the one of worker III is completed at 571 episodes. Therefore, we can know that the learning speed becomes much higher and the variation in learning time for each RIP device is also reduced when the proposed scheme is applied.

It is also noted in Figure 5b that the score value of Worker I and III increases suddenly at 287 episodes. It happens because Worker II transfers its mature actor model into Worker I and III. The model transfer plays a large role in shortening the learning time.

However, we can also know that the additional learning at Worker I and III are still needed over several episodes to complete the learning process, even though the mature model of Worker II is transferred to them. To find out the reason for additional learning, another experiment is conducted. We apply the exact same amount of force to each of three RIP devices (i.e., Quanser QUBE^TM^-Servo 2) 100 times in different directions, and measure the Pearson correlation between the changes of the motor and the pendulum angles for the three devices. Pearson correlation [33] is commonly used to find the relationship between two random variables. The Pearson correlation coefficient has +1 if the two variables X and Y are exactly the same, 0 if they are completely different, and −1 if they are exactly the same in the opposite direction. Table 2 shows the results of the homogeneity test for the dynamics of three RIP devices of the same type. As known from the two tables, the angles of motor and pendulum are changed differently even though the forces applied in different directions are constant over 100 times. For each RIP device, in particular, the change in motor angle is more varied than the change in the pendulum angle. For multiple RIP devices of the same type, as a result, their dynamics are slightly different from each other, even though they are produced on the same manufacturing line. This means that the additional learning at Worker I and III is still needed even after receiving the mature model of Worker II.

Figure 6 shows the efficiency of the federated reinforcement learning according to the number of workers. For two workers and three workers, the experiments are conducted five times and the average score per episode is shown in the figure. As shown in the figure, the higher the number of workers, the faster the learning process. In the proposed scheme, each worker share its learning experiences, i.e., the gradient of loss function computed on each RIP device, with each other. Each agent also transfers its actor model parameters into other agents when its learning process is finished. As the number of workers increases, the effects of such two strategies increase.

In our previous study [34], we validated the proposed federated reinforcement learning through the CartPole simulation software environment of the OpenAI Gym. We also found that the performance increases as the number of workers increases from one to eight in the simulation environment. The source code of the experiments is publicly available at https://github.com/glenn89/FederatedRL (Appendix A).

## 6. Conclusions

In this work, we have shown that the proposed federated reinforcement learning scheme can successfully control multiple real IoT devices of the same type but with slightly different dynamics. We adopted Actor–Critic PPO as a reinforcement learning algorithm and applied it to the federated learning architecture. The proposed approach includes the gradient sharing and model transfer methods to facilitate the learning process, and it turned out that they can expedite the learning process by about 1.5 times. We also have shown that learning is further accelerated by increasing the number of workers. Learning speed was improved by about 38% when three workers were used compared to two workers. As the future work, we plan to study the system heterogeneity in the federated reinforcement learning architecture. The computational and communication capabilities of each IoT device in federated networks may differ. Moreover, some IoT devices may also be unreliable, and it is common for an IoT device to drop out at a given round. In these heterogeneous settings, how to expedite the reinforcement learning process will be studied.

## Figures and Tables

**Figure 1 sensors-20-01359-f001:**
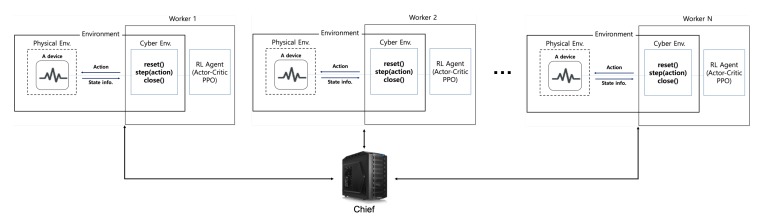
The system configuration for the proposed federated reinforcement learning.

**Figure 2 sensors-20-01359-f002:**
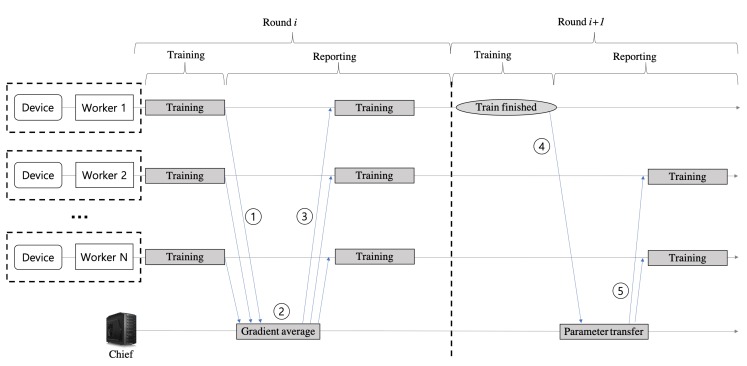
The federated reinforcement learning overall procedure.

**Figure 3 sensors-20-01359-f003:**
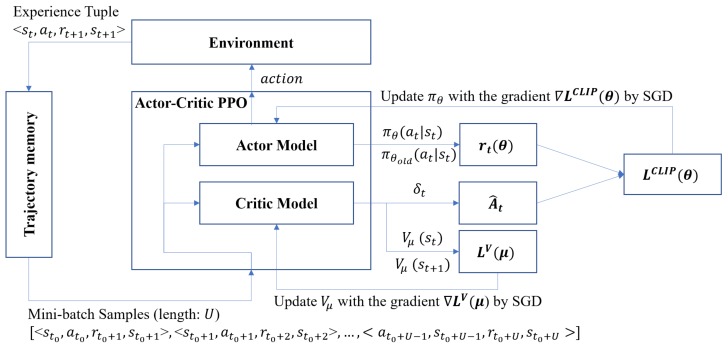
The actor–critic proximal policy optimization (Actor–Critic PPO) algorithm process.

**Figure 4 sensors-20-01359-f004:**
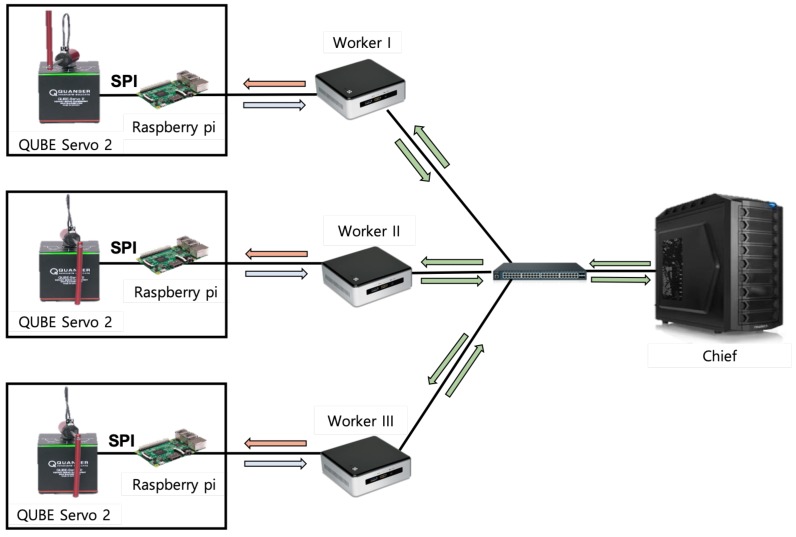
Our experiment configuration with multiple rotary inverted pendulum (RIP) devices, multiple workers, and one chief.

**Figure 5 sensors-20-01359-f005:**
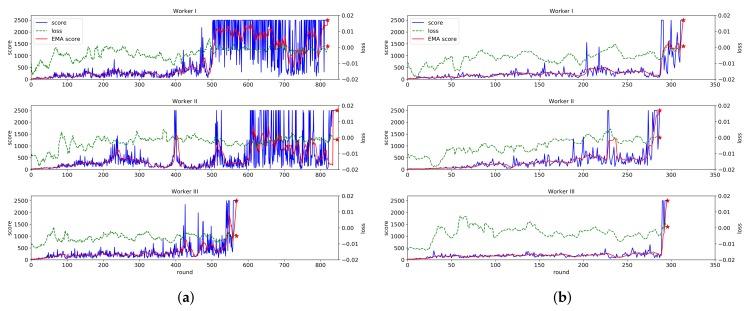
Effectiveness of the proposed federated reinforcement learning scheme. The blue line represents the score for each round, and the red line represents the weighted moving average (WMA) of the scores from the last 10 rounds. The green dotted line indicates the loss value for each round. (**a**) Change of score and loss values without the proposed scheme. (**b**) Change of score and loss values with the proposed scheme.

**Figure 6 sensors-20-01359-f006:**
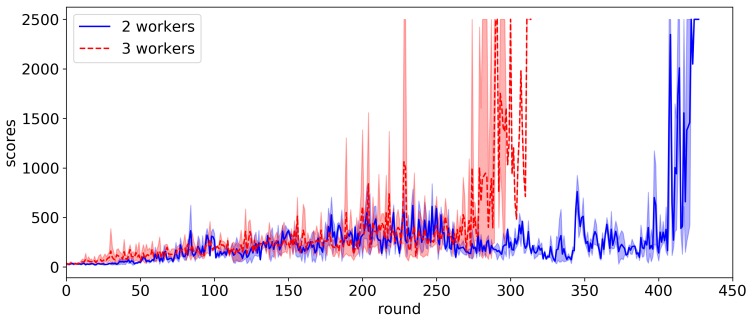
Comparison of learning speed in terms of the number of workers.

**Table 1 sensors-20-01359-t001:** Hyper-parameter configuration for Actor–Critic PPO.

Hyper-parameter	Value
Clipping parameter (ϵ)	0.9
Model optimization algorithm	Adam
GAE parameter (λ)	0.99
Learning rate for the critic model ($\eta_{\mu }$)	0.001
Learning rate for the actor model ($\eta_{\theta }$)	0.001
Trajectory memory size	200
Batch size (*U*)	64
Number of model optimizations in one round (*K*)	10

**Table 2 sensors-20-01359-t002:** Results of homogeneity test for dynamics of multiple IoT devices of the same type.

(a) Pearson correlation matrix of motor angle changes			
**Motor Angle**	**RIP I**	**RIP II**	**RIP III**
RIP I	1	0.77	0.86
RIP II	-	1	0.75
RIP III	-	-	1
**(b) Pearson correlation matrix of pendulum angle changes**			
**Pendulum Angle**	**RIP I**	**RIP II**	**RIP III**
RIP I	1	0.98	0.96
RIP II	-	1	0.98
RIP III	-	-	1

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
