# Peer review of "Federated Reinforcement Learning for Training Control Policies on Multiple IoT Devices"

_sensors, 2020, doi:10.3390/s20051359_

Round 1

Reviewer 1 Report

The manuscript describes an application of federated learning on IoT devices. The application is the inverted pendulum, is a classical example of a control problem.

Overall, I have few suggestions for improvement of the manuscript. The difference from the state of the art is sufficiently explained. The algorithms are clearly presented. The experiments are done on real devices, the methodology is sound, and the demonstrated improvement from authors' method is convincing.

I suggest the authors proofread the manuscript and check for typos - e.g. line 96 "eznvironment".

The authors should provide some numerical results in the conclusion.

They should also qualify the statement that learning is facilitated by increasing the number of worker. Given that they only do a single experiment where the learning rate with two and three workers are quantified, the experimental evidence provided is not sufficient to make such a general claim."

Author Response

[Answers for Reviewer 1’ Comments]

First of all, we would like to thank you for all that you commented for our paper.

*** Reviewer 1 ***

The manuscript describes an application of federated learning on IoT devices. The application is the inverted pendulum, is a classical example of a control problem.

Overall, I have few suggestions for improvement of the manuscript. The difference from the state of the art is sufficiently explained. The algorithms are clearly presented. The experiments are done on real devices, the methodology is sound, and the demonstrated improvement from authors' method is convincing.

Point 1. I suggest the authors proofread the manuscript and check for typos - e.g. line 96 "eznvironment".

[Response 1]

As you pointed out, we changed "eznvironment" into environment the line 96, and the overall English sentences polishing was again done to improve the quality of the revised paper.

Point 2. The authors should provide some numerical results in the conclusion.

[Response 2]

We wrote down additional numerical results in the conclusion section. The added contents are as follows:

“The proposed approach includes the gradient sharing and model transfer methods to facilitate the learning process, and it turned out that they can expedite the learning process by about 1.5 times. We also have shown that learning is further accelerated by increasing the number of workers. Learning speed was improved by about 38% when three workers were used compared to two workers.”

Point 3. They should also qualify the statement that learning is facilitated by increasing the number of worker. Given that they only do a single experiment where the learning rate with two and three workers are quantified, the experimental evidence provided is not sufficient to make such a general claim."

[Response 3]

We agree that we need to increase the number of workers. But we only have three Qube-Servo 2 in our laboratory, and the devices are very expensive. Just before writing this paper, we validated the proposed federated reinforcement learning through the CartPole simulation software environment of the OpenAI Gym. In the previous studies, we found that the performance increases as the number of workers increases from one to eight in the simulation environment. The paper related to the previous research is the following Korean paper.

Lim, H.K.; Kim, J.B.; Han, Y.H.  Learning Performance Improvement Using Federated Reinforcement Learning Based on Distributed Multi-Agents. KICS Fall Conference, 2019, pp. 293–29

The following content was also added into the last paragraph of the Section 5.3.

“In our previous study [34], we validated the proposed federated reinforcement learning through the CartPole simulation software environment of the OpenAI Gym. We also found that the performance increases as the number of workers increases from one to eight in the simulation environment.”

Thanks for your comments.

Reviewer 2 Report

  • The paper presents a federated reinforcement learning scheme and describes its architecture, the applied methods and algorithms, and also the obtained results after applying the scheme to real devices.
  • The paper clearly identifies the scope and the contribution. It highlights the results and experiments. The authors give enough information for further implementation of their proposed scheme.
  • It would be useful to provide some Python pieces of code that show different aspects of the implementation.
  • There are minor English misspellings or unclear, repetitive words used in some expressions. See line 96 or 167, for example.

Author Response

[Answers for Reviewer 2’ Comments]

First of all, we would like to thank you for all that you commented for our paper.

*** Reviewer 2 ***

The paper presents a federated reinforcement learning scheme and describes its architecture, the applied methods and algorithms, and also the obtained results after applying the scheme to real devices.

The paper clearly identifies the scope and the contribution. It highlights the results and experiments. The authors give enough information for further implementation of their proposed scheme.

Point 1. It would be useful to provide some Python pieces of code that show different aspects of the implementation.

[Response 1]

Please understand that the Python code for the proposed federated reinforcement learning is very long. In particular, there are two important parts of codes, each supporting the chief and the workers, respectively. If we insert the Python code into the paper, the paper will become long. So, we just publish our code on Github (https://github.com/glenn89/FederatedRL) publicly.

After Section 6 Conclusion, we mentioned the Github URL into Supplementary Material.

Point 2. There are minor English misspellings or unclear, repetitive words used in some expressions. See line 96 or 167, for example.

[Response 2]

As you pointed out, we changed "eznvironment" into "environment" in the line 96, and removed the duplicated word "the" in the line 167. The overall English sentences polishing was again done to improve the quality of the revised paper.

Thanks for your comments.
